# Following the COVID-19 Experience, Many Patients with Type 1 Diabetes Wish to Use Telemedicine in a Hybrid Format

**DOI:** 10.3390/ijerph182111309

**Published:** 2021-10-28

**Authors:** Tal Schiller, Taiba Zornitzki, Viviana Ostrovsky, Danielle Sapojnik, Lee Cohen, Tamila Kunyavski, Hilla Knobler, Alena Kirzhner

**Affiliations:** Department of Endocrinology, Diabetes and Metabolic Disease, Kaplan Medical Center and Faculty of Medicine, Hebrew University of Jerusalem, Jerusalem 9190401, Israel; schillerta@gmail.com (T.S.); lior_zo@inter.net.il (T.Z.); viviana.ostrovsky@gmail.com (V.O.); daniellesa1@gmail.com (D.S.); lee14281@gmail.com (L.C.); tamila4ka@gmail.com (T.K.); Hilla.Knobler@gmail.com (H.K.)

**Keywords:** telemedicine, type 1 diabetes, remote consultation, virtual medicine

## Abstract

Background: The COVID-19 pandemic has brought to light both challenges and unique opportunities regarding type 1 diabetes (T1D) management, including the usage of telemedicine platforms. Methods: This study was conducted in a tertiary hospital diabetes clinic. All consecutive T1D patients during March and June 2021 were asked to fill out a structured anonymous questionnaire that aimed to determine their preference regarding continuous use of a virtual platform. Results: In total, 126 T1D patients answered the questionnaire, of whom 51% were under the age of 40, half were men, half used insulin pumps, and 69% used continuous glucose monitoring. During the pandemic, the exposure of patients to virtual visits has grown about twofold, from 29% to 53%. Of the respondents, 49% expressed an interest in future usage of a virtual platform, but most of them preferred use in a hybrid manner. We found an association between preference to use telemedicine in the future and younger age, previous virtual platform experience, and confidence in being able to download data. Conclusions: Our data demonstrate that the COVID-19 experience has led to a growing interest of T1D patients in using the hybrid format of telemedicine. However, we still need to better understand who will benefit most from this platform and assess its cost-effectiveness and organization.

## 1. Introduction

The COVID-19 pandemic has dramatically influenced many aspects of patients’ treatment. The Israeli government implemented several extended lockdown periods starting from April 2020 through March 2021, and the healthcare system had to adapt. During the first lockdown, all non-urgent activities were cancelled, and outpatient services were dramatically reduced. However, during the following lockdowns, outpatient services resumed most of their usual activity, and patients were allowed to choose how to conduct visits. Israel was the first to implement mass vaccination, starting on 20 December 2020, and lockdown subsequently ended in March 2021. Following the vaccination program, the rate of COVID-19 infection dropped dramatically.

During the past years, many T1D patients in our diabetic clinic have been using phone calls or text messages via WhatsApp in parallel with in-person visits. Clinic appointments are flexible and scheduled according to the patients’ needs and can range from every week in newly diagnosed patients to once yearly in stable and well-controlled patients. The average time per appointment is 30–60 min. As mentioned, during the first lockdown, in-person visits were replaced almost entirely by phone calls. Professionals of all disciplines in the clinic, including nurses, physicians, and dietitians, could conduct visits either virtually or in-person. Excluded were newly diagnosed T1D patients and patients new to the clinic, for whom the staff decided that there was a need for at least one in-person visit.

Over the past years, the usage of telemedicine has generated global interest, as evidenced by the increasing number of studies published on this topic [1,2,3,4]. 

Telemedicine has been proven to be very effective in remote areas with poor health facilities or access limitations [2,3]. According to these studies, some medical specialists, such as those in radiology, dermatology, pathology, and psychiatry, have used telemedicine more frequently than others [4]. The most common purposes of telemedicine use were clinical care, follow-up, and medical education. On the other hand, there are issues that merit consideration, such as the importance of providing telemedicine service according to government legislation; ensuring confidentiality, integrity, and availability of personal and health data through electronic tools; and adhering to ethical and legal standards [2,3]. 

Through the accumulating experience of many diabetes clinics, it was shown that it is possible to provide good-quality care through telemedicine: to maintain or even improve glycemic control, improve self-management and life-quality [5], and reduce diabetes-related distress among young people with T1D [6]. 

Nevertheless, healthcare professionals express varying opinions regarding the use of telemedicine. Despite acknowledging the positive aspects, some are worried about the need to preserve the in-person connection between patients and healthcare providers and the inability to perform a physical examination [7,8]. In a study from Italy using the Patients Assessment Chronic Illness Questionnaire in insulin pump users, it was demonstrated that health professionals focused mainly on clinical aspects of telemedicine with patients’ satisfaction and less on social aspects [9]. 

There are limited data regarding the short-term usage of telemedicine during the COVID-19 pandemic [10,11,12,13,14]. After the first COVID-19 lockdown, we conducted a questionnaire survey aimed to assess the needs of T1DM patients regarding virtual care [15]. We concluded that patients were not ready to give up in-person visits and that older populations needed further guidance and education. However, there is still a need to evaluate the long-term perception of telemedicine in the treatment of T1D. The unique situation in Israel of mass immunization and reduced COVID-19 infection rate enabled us to address this question. The current study was designed to understand patients’ attitudes a year into the pandemic and whether patients wish to use telemedicine in the long term. 

The aims of this follow up work were: (1) to determine how many of the patients are interested in long-term virtual care and, specifically, exclusive virtual care or a hybrid modality (combining in-person and virtual visits); (2) to characterize the patients who prefer each modality; and (3) to determine whether immunization alters the patient’s preference.

## 2. Methods

### 2.1. Study Design

The study was conducted between March and June 2021 at the Diabetes Clinic at Kaplan Medical Center, a university-affiliated hospital serving as a referral center for patients living in the central–southern region of Israel. Patients are treated by a multidisciplinary team including physicians, nurses, dietitians, and psychologists. This study is a continuation of the study carried out in our clinic during August to October 2020 between the first and second lockdowns in Israel, but on a different cohort. All consecutive adult (age ≥18 years) T1D patients were asked to fill out a structured questionnaire while attending the clinic or via WhatsApp. We excluded patients who (1) had new-onset T1D; (2) were attending the clinic for the first time; or (3) were unable to complete the questionnaire due to a language barrier.

The questionnaire was designed by the Diabetes Clinic team and was anonymous. The questionnaire and the study protocol were approved by the local Institutional Review Board. 

The questions focused on whether patients wish to use a virtual platform alone or in a hybrid manner and included demographic data, experience of virtual medicine before and during the COVID-19 pandemic, perception of visits, and future preference. 

### 2.2. Statistical Analysis 

Data are presented as the mean ± standard deviation, median (IQR), or percentage. Continuous variables in the various study groups were tested for normality by the Shapiro–Wilk test, and when a non-normal distribution was found, nonparametric tests were performed. The Mann–Whitney test was performed to compare two groups. When the distribution was normal, the *t*-test was used. The Chi-square test was used to assess the relationship between two categorical variables. A *p*-value of <0.05 was considered statistically significant. Data were analyzed using SPSS25.

## 3. Results

The baseline characteristics of the 126 participants who answered the questionnaire are presented in Table 1. 

Half of our patients were men and 51% were under the age of 40, while 54% used insulin pumps and 69% used continuous glucose monitoring. More than one-third of our patients had HbA_1C_ less than 7%, and 22% had HbA_1C_ greater than 9%. Eight were pregnant during the study period. Out of these eight women, five wished to use a hybrid visit format and three preferred in-person visits only. In the year prior to the study, 34 patients visited the clinic more than four times; 22 patients visited the clinic three times; 23 patients visited two times; and 47 patients had one visit. The frequency of visits was not influenced by the mode of visit, and the number of visits did not influence patient preference.

As shown in Table 2, during the pandemic, the exposure of patients to virtual visits grew by about twofold, from 29% to 53%. Most virtual visits were conducted as part of a routine follow-up, and only a minority were due to an urgent condition such as persistent hyperglycemia or following a significant hypoglycemia event. Out of the 126 patients, 49% wished to use the virtual platform in the future, and 42% of the whole group wished to do so in a hybrid manner (Figure 1). Only 7% wished to perform only virtual visits. 

Among the perceived advantages of in-person visits, nearly 70% of respondents indicated better communication, 42% believed that they provide better quality of care, and 66% considered the physical examination as an important component of the visits. At the same time, transportation and parking issues, work time loss, and overall time loss were indicated as limitations of in-person visits by 31%, 18%, and 18% of respondents, respectively. Forty-seven percent answered that fear of COVID-19 transmission is a limitation for attending the clinic. However, only 28% reported that immunization changed their willingness to visit the clinic.

Lastly, we wanted to see whether baseline characteristics predict the willingness of patients to use telemedicine in the future. As presented in Table 3, patients who were interested in future use of telemedicine were younger, more of them had previous virtual platform experience, and they were more confident in downloading data at home. In contrast, sex, education, diabetes duration, mode of insulin treatment, distance from the clinic, and HbA_1C_ were not associated with willingness to use telemedicine in the future. We also analyzed according to HbA1_C_ below and above 7% (*p* = 0.289); HbA1_C_ below and above 8% (*p* = 0.08); and HbA1_C_ below 7%, as compared to above 9% (*p* = 0.104).

## 4. Discussion

In the current study, we found that 49% of our patients wish to use telemedicine as part of their long-term treatment. However, most preferred it to be in a hybrid manner. Younger age, former experience with telemedicine, and confidence in being able to download data at home are associated with patients’ preference to use telemedicine. The finding that the younger population with better technological abilities prefers telemedicine use is not surprising. Perceived gains include time and cost savings, while obstacles include more limited communication and the lack of a physical examination. Other perceived obstacles include the patient’s ability to download data and data safety concerns.

### 4.1. Telemedicine and Routine Glycemic Control 

Despite the isolation and lifestyle changes, recent published studies did not demonstrate, in general, that T1D patients experienced deterioration in their glucose control during the COVID-19 lockdowns [16,17,18,19,20]. 

Predieri et al. conducted an observational study on 62 children and adolescents with T1D using the continuous glucose monitoring (CGM) Dexcom G6. Ambulatory glucose profile (AGP) data from the three months before lockdown and from three months of consecutive lockdown were compared. They observed that glucose control improved with increased Time in Range (TIR), and the median value of the glucose management indicator (GMI) decreased from 7.4% to 7.25% at the end of the lockdown frame [21].

Parise et al. analyzed glucose control among 166 adult patients affected by T1D who completed two virtual visits during the lockdown period. TIR significantly increased from the first to the second virtual visit. This increase was more marked among patients using the traditional Glucometer than among those using CGM, and also more marked in those with a baseline GMI of ≥7.5% than in those with a GMI of <7.5% [22]. However, these encouraging findings may not represent the situation in the general T1D population treated by less-experienced staff and or in less-equipped patients. 

### 4.2. Telemedicine in New-Onset Type 1 and Urgent Care 

Recent retrospective case studies reported that telemedicine can be used effectively by multidisciplinary teams for new-onset T1D and that remote treatment of ketoacidosis prevented hospital admissions [23,24]. 

The results of these studies are encouraging, but further work is required to evaluate long-term sustainability and support.

### 4.3. The Perception of Telemedicine by T1D Patients and by Healthcare Providers 

Information regarding the attitude of T1D patients towards the usage of telemedicine is limited. The largest study that evaluated the perception of telemedicine by patients with T1D was published by Scott et al. [13]. This study demonstrated a positive attitude of patients with T1D towards remote medicine. Seventy-five percent of patients stated that they would consider remote visits beyond the COVID-19 pandemic. A smaller study from Australia also found that most T1D patients regarded their two-week exposure to telemedicine as a positive experience [14]. Positive aspects included time and money savings, but lack of a physical examination was perceived as a major drawback. Positive experience with telemedicine was noted among adolescents and parents in a small study by Lim et al. [12]. Twenty-eight patients attended a structured transition education program, aimed to address the unique challenges faced by patients with T1D. Both adolescents and parents reported that telemedicine was similar to in-person visits, and 20% thought it even to be superior to the in-person visits [12].

In our previous study after the first lockdown, we found that most patients preferred a combination of in-person and virtual visits [15]. The patients’ perception was that in-person communication offers better communication with the medical team and better-quality care, but almost all patients considered virtual visits to be time saving. About one-quarter of the patients expected virtual visits to improve their glycemic control, and a minority emphasized the advantage of it being less expensive.

### 4.4. Where Does Telemedicine Stand Today?

Although telemedicine has gained popularity during the COVID-19 pandemic, previous studies assessed the use of telemedicine in the short term, and there are limited data regarding its long-term usage. A major issue that has to be considered is the patients’ willingness to continue using telemedicine for long periods of time. In the current study, we assessed patients’ preference a year into the pandemic after experiencing the different treatment modalities. 

The majority of the patients in the current study gained experience in virtual care, and most of them agree that telemedicine is convenient. However, virtual consultations are not always suitable, and there are circumstances in which in-person contact between patients and healthcare providers is invaluable [25]. 

Virtual clinics increase the opportunities for people with diabetes to retain contact with the diabetic clinic staff. This can allow better continuity of care and provide a sense of security for many patients since they can communicate from their home environment if needed [26]. However, there are still no clear guidelines on how to organize the work of the multiprofessional team and provide comprehensive treatment to the growing number of patients. Providing telemedicine requires further professional abilities in the management of T1D. The rapid development of technology places higher demands on teamwork to provide care, education, and support [27]. 

For those who wish to continue using telemedicine either exclusively or in a hybrid format, there is a need to establish methods of keeping proper and safe data records in order not to lose vital information. 

For those who are reluctant to use telemedicine, other questions rise: Do we use solely in-person visits? What about those who are reluctant to visit the clinic but are not capable of downloading data or have data safety concerns? How do we manage downloading of data in the older populations and those who are less technologically adept? 

Other issues such as cost effectiveness and reimbursement, choice of the method and device for the virtual visit, multidisciplinary team involvement, visit duration, and visit intervals are still to be answered for effective work planning, with special emphasis on choosing the right patients for whom telemedicine may be useful [28].

The strengths of our study include the following: (1) a comprehensive questionnaire with detailed visit modalities; (2) a cohort of T1D patients with a wide range of age groups, HbA_1C_, education levels, durations of diabetes, and treatment modalities; (3) patients were able to answer the questionnaires either during the clinic visit or via WhatsApp, thus enabling receipt of data not only from patients with internet access; and (4) a significant proportion of our patients already had experience with virtual medicine in the past and during the pandemic. The limitations include the following: (1) the patients included are treated in a hospital clinic and may not be representative of patients treated in the community; (2) not all patients who answered the questionnaire had experienced virtual medicine, despite the explanations received from the diabetes clinic team; and (3) all survey data were self-reported, so reporting accuracy could be a concern.

## 5. Conclusions

The COVID-19 pandemic poses an unprecedented challenge to patients with chronic disease, including diabetes. At the same time, it has led to invaluable contributions to further planning and development of care. We need to develop a better understanding of who may benefit versus possible caveats and losses in the continuous use of telemedicine. The current study data suggest that a hybrid format of visits is preferred by most patients. In addition, older patients may need different resources and guidance to implement telemedicine. Further studies are needed to enable stratification of those patients who will benefit the most.

## Figures and Tables

**Figure 1 ijerph-18-11309-f001:**
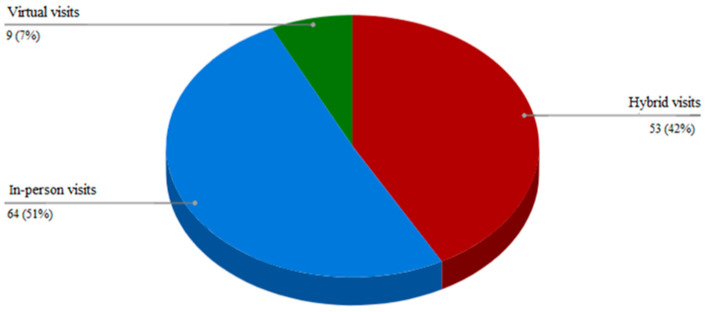
Preferred visit mode.

**Table 1 ijerph-18-11309-t001:** Baseline characteristics of the study population (N = 126).

Variable	N (%)
**Males/Females**	63/63
**Age, years**	
18–40	64 (51)
41–60	35 (28)
Over 60	27 (21)
**Education, years**	
≥12	70 (56)
<12	56 (44)
**Diabetes duration, years**	
>20	51 (40)
10–20	46 (37)
<10	29 (23)
**HbA_1C_%, mmol/mol**	
<7 (53 mmol/mol)	46 (37)
7.1–8.0 (53–64 mmol/mol)	37 (29)
8.1–9.0 (65–75 mmol/mol)	15 (12)
>9 (75 mmol/mol)	28 (22)
**Mode of treatment**	
Insulin pump	68 (54)
Multiple daily injections (MDI)	58 (46)
Continuous glucose monitoring (CGM)	87 (69)
**Estimated travel time to the clinic, minutes**	
<30	76 (60)
30–60	43 (34)
>60	7 (6)
**COVID-19 status ***	
2 vaccinations	93 (74)
1 vaccination	9 (7)
Not vaccinated	14 (11)
Recovered	9 (7)

* 1—missing data.

**Table 2 ijerph-18-11309-t002:** Patients’ perceptions of virtual vs. in-person visits (N = 126).

Variable	N (%)
**Telemedicine experience before COVID-19 pandemic**	
No	89 (71)
Yes	37 (29)
**Telemedicine experience starting during the COVID-19 pandemic**	
No	59 (47)
Yes	67 (53)
**Willingness to use virtual medicine post-COVID-19 pandemic**	
No	64 (51)
Yes	62 (49)
**Perceived limitations of in-person visits**	
Loss of working hours	22 (18)
Transportation and parking issues	39 (31)
Time-consuming	22 (18)
Concern over COVID-19 infection	59 (47)
**Perceived advantages of in-person visits**	
Better communication with the medical team	89 (71)
Better care quality	53 (42)
Better responsiveness to treatment	53 (42)
Possibility of physical examination	83 (66)
**The level of confidence in being able to download data at home**	
Confident	75 (60)
Partly confident	31 (25)
Not confident at all	20 (15)
**Does data safety prevent you from data downloading?**	
No	110 (87)
Yes	16 (13)
**Does the COVID-19 vaccination affect your decision to attend the clinic?**	
No	91 (72)
Yes	35 (28)

**Table 3 ijerph-18-11309-t003:** Comparison of baseline characteristics between patients who wish to use telemedicine in the future and those who do not.

	Interested in Future Use of Virtual Medicine (N = 62)	Not Interested in Future Use of Virtual Medicine (N = 64)	*p*-Value
**Age, years**			
18–40	37 (60)	27 (42)	**0.049 ***
41–60	17 (27)	18 (28)	
>60	8 (13)	19 (30)	
**Sex:**			
Males	31 (50)	32 (50)	1.000
Females	31 (50)	32 (50)	
**Travel time, minutes**			
<30	38 (61)	38 (59)	0.179
30–60	18 (29)	25 (39)	
>60	6 (10)	1 (2)	
**Education, years**			
≥12	37 (60)	33 52)	0.461
<12	25 (40)	31 (48)	
**Diabetes duration, years**			
>20	20 (32)	31 (48)	0.103
10–20	28 (45)	18 (28)	
<10	14 (23)	15 (24)	
**Treatment type:**			
Pump	37 (60)	31 (48)	0.277
MDI	25 (40)	33 (52)	
Glucometer	19 (31)	20 (31)	0.352
FGM	14 (23)	21 (33)	
CGM	29 (47)	23 (36)	
**HbA1_C_% (mmol/mol)**			
<7 (53 mmol/mol)	26 (43)	20 (31)	0.184
7.1–8.0 (53–64 mmol/mol)	20 (32)	17 (27)	
8.1–9.0 (65–75 mmol/mol)	4 (6)	11 (17)	
>9 (75 mmol/mol)	12 (19)	16 (25)	
**Virtual medicine experience prior to COVID-19**			
No	44 (71)	45 (70)	1.000
Yes	18 (29)	19 (30)	
**Virtual medicine experience starting at COVID-19 period**			
No	22 (35)	37 (58)	**0.02 ***
Yes	40 (65)	27 (42)	
**Level of confidence in being able to download data at home**			
Confident	47 (76)	28 (44)	**<0.001 ***
Partly confident	11 (18)	20 (31)	
Not confident at all	4 (6)	16 (25)	

CGM—continuous glucose monitoring, FGM—flash glucose monitoring, MDI—multiple daily injections. * Statistically significant *p* < 0.005.

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
