# Peer review of "Following the COVID-19 Experience, Many Patients with Type 1 Diabetes Wish to Use Telemedicine in a Hybrid Format"

_ijerph, 2021, doi:10.3390/ijerph182111309_

Round 1
Reviewer 1 Report
The authors presented a very interesting work on the current problem of outpatient opiates in people with type 1 diabetes. The study group is a cross-section of different patients, which is an advantage of this work. He is an important voice in the discussion on structuring and formalizing the issue of teleportation in diabetology. I would have a few comments: 1. Please add information about the situation in Israel - how often a person with diabetes has an appointment at the clinic (how many / year), how long does the standard visit last? 2. Please explain in the methodology what a hybrid visit is 3. How many visits did each of the respondents make during the study, were they more frequent than usual with teleporadach? 4. Will only the doctor or the nurse / educator take part in a teleport and hybrid visit? 5. The chapter should be shortened - results, the content that is presented in the table or graph does not require detailed description in the text. 6. What percentage of people who prefer virtual visits prefer them in a hybrid form (the results only show the percentage of the entire group, why?) 7. Was there a comparison of data such as age, HbA1c, duration, not broken down into subgroups but as mean or median values? The statistics would be stronger using Mann-Whitney or T-student. The methods describe that they were used, but you don't see them in the results. 8. Was there an attempt to divide the group in terms of metabolic control into 2 groups - well and badly balanced, and were there differences in the context of the type of visits? Maybe people who don't really care about good control will avoid live visits to hide things easier? 9. Were there pregnant women in the study group? 10. Were there any cases of DKA or hospitalization during the follow-up (after visiting the clinic and qualifying for the study?) 11. Was there a change in the alignment for better or worse after the examination depending on the type of visits received? 12. The results were stronger if the study group was larger, it may be worth extending the study if the questionnaire is on-line.
Reviewer 2 Report
Dear Authors,
This paper concerned interesting problem of changes in diabetes care in COVID era. This is interesting and potentially important topic, but if it will be informatively constructed study. In Your study the aims is concerned only patients willing’s in small group of patients from hospital affiliated ambulatory clinic. Telemedicine is new form of diabetes care, and has increasing significance, especially in pandemic era, and era of diabetes pandemia. We have no option, only adopt telemedicine to diabetes care, especially because it is one of the most suitable disease, possible to remote control. As in Your centre, everywhere it should be care using telemedicine additionally to regular visits, or hybrid. Our aim should be the question how to build the care using telemedicine, not who like it, how teach patients using it, not who use it willingly, how to organize it to get the best results, and how many personal visits in the clinic per year is absolutely needed. Additionally, I have a few questions and comments:
- What is consisted of visit by the whatsapp? It was only by whatsapp or by phone, and transfer of data was using whatsapp?
- In the table no 3, which parameters were analysed- if there were “interested” vs “not interested” why You took only one analysis (one p value), and which analysis it concerns. If it was analysis of 3 groups, it should be used the other statistic test- which is not mentioned in statistical analysis.
- What for was analysed the method of questionnaire filling? This is not important at all, and for the aim of the study.
- In the discussion it was not taken in consideration that glycaemic control during lockdown was influenced by the fact that patients, sometimes for the first time in life, have more time at home, for regular control, regular meals, and children with diabetes were on much closer parent’s control. After a period of confinement in the houses the effects of physical inactivity appeared.
Round 2
Reviewer 2 Report
Authors reply for my particular comments but not for the meritum. I think that idea of this study was not good. In my opinion this is not innovative and not important.
